# A Disease-Prediction Protocol Integrating Triage Priority and BERT-Based Transfer Learning for Intelligent Triage

**DOI:** 10.3390/bioengineering10040420

**Published:** 2023-03-27

**Authors:** Boran Wang, Zhuliang Gao, Zhikang Lin, Rui Wang

**Affiliations:** 1School of Computer and Communication Engineering, University of Science and Technology Beijing (USTB), Beijing 100083, China; 2Beijing Tiantan Hospital, Capital Medical University, Beijing 100070, China; 3Shunde Graduate School, University of Science and Technology Beijing, Foshan 528300, China

**Keywords:** intelligent triage, transfer learning, neurological, multilabel, BERT, cost-sensitive, data imbalance

## Abstract

Large hospitals can be complex, with numerous discipline and subspecialty settings. Patients may have limited medical knowledge, making it difficult for them to determine which department to visit. As a result, visits to the wrong departments and unnecessary appointments are common. To address this issue, modern hospitals require a remote system capable of performing intelligent triage, enabling patients to perform self-service triage. To address the challenges outlined above, this study presents an intelligent triage system based on transfer learning, capable of processing multilabel neurological medical texts. The system predicts a diagnosis and corresponding department based on the patient’s input. It utilizes the triage priority (TP) method to label diagnostic combinations found in medical records, converting a multilabel problem into a single-label one. The system considers disease severity and reduces the “class overlapping” of the dataset. The BERT model classifies the chief complaint text, predicting a primary diagnosis corresponding to the complaint. To address data imbalance, a composite loss function based on cost-sensitive learning is added to the BERT architecture. The study results indicate that the TP method achieves a classification accuracy of 87.47% on medical record text, outperforming other problem transformation methods. By incorporating the composite loss function, the system’s accuracy rate improves to 88.38% surpassing other loss functions. Compared to traditional methods, this system does not introduce significant complexity, yet substantially improves triage accuracy, reduces patient input confusion, and enhances hospital triage capabilities, ultimately improving the patient’s medical experience. The findings could provide a reference for intelligent triage development.

## 1. Introduction

Hospitals need a safe, good, and mature intelligent triage system to help patients choose the right department for treatment [1]. Such a triage system allows the hospital to quickly focus on collecting data, and data analysis, judgment, classification, and division [2]. Due to the lack of medical knowledge reserve of the patient population, the corresponding relationship between the discipline setting of large hospitals and clinical diseases is more complicated [3], and it is often impossible to quickly locate the outpatient department of symptomatic treatment, and there are more ineffective visits [4]. Medical staff, such as triage nurses, can help some patients, but due to the large population base of patients, problems such as triage inefficiency still need to be solved. The traditional triage method has been unable to fully meet the needs of patients.

Based on literature searches, it is clear that many scholars have studied machine learning-based development methods for building intelligent triage systems or medical text-classification systems [5]. The conventional triage mode is based on the experience of the triage personnel. The experience of the triage personnel is the standard of the triage. Each hospital lacks a unified preexamination triage standard and specific implementation rules. The application of artificial intelligence in the development of triage is conducive to the rapid interpretation of clinical data and the classification and diagnosis of patients’ conditions. Researchers have developed many prescreening and triage tools, such as the START system based on logistic regression [6] and the syndrome-monitoring system based on text classifier [7].

With the rapid development of neural networks, there have been some methods based on deep learning for triage. Gligorijevic et al. [8] proposed a deep model based on word attention mechanism, which combines structured and unstructured data to predict the resources that emergency patients may need. The accuracy of the model for multiclassification tasks reaches 44%. The neural network for text classification tasks can be extended to intelligent triage. Lv et al. [9] improved CNN and established an interactive intelligent medical prediction and evaluation model based on a deep learning model. Gao [10] uses CNN, RNN, and RCNN for clinical department recommendation, and compares the results of the three. The experimental results show that the accuracy of RCNN is higher than that of CNN and RNN, which can reach 76.5%. However, most of the methods and studies at this stage solve the triage of general practice, and there is no relevant research on a more accurate subspecialty. In the field of neurosurgery, there is no reported auxiliary triage system. The types of nervous system diseases are complex and diverse, and the disease changes rapidly. The disease often involves multiple diagnoses. The similarity between the medical records of each diagnosis is high, which causes great difficulties for triage. In addition, the training period of neurosurgery triage nurses is long, the cost is high, the medical level is uneven, the rate of withdrawal and replacement is high, and the auxiliary triage system is more needed than other subspecialties.

Taking T hospital (alias) as an example, the hospital is mainly based on neurology and neurosurgery. Neurology includes vascular neurology, neurocritical medicine, dyskinesia, epilepsy, nerve infection, and immunology, neuromuscular disease, cognitive impairment, and headache.

Neurosurgery covers 21 wards, including cerebrovascular disease, pediatric neurosurgery, craniocerebral trauma, spinal cord, supratentorial tumor, infratentorial tumor, skull base brain stem, functional neurosurgery, neuroendoscopy, intracranial and extracranial communication tumor, peripheral neurosurgery, comprehensive treatment of nerve tumor, etc. The existing triage system in China cannot meet the requirements of patients, and it is very difficult to accurately register patients with insufficient medical knowledge.

This paper proposes a deep learning-based intelligent triage system (TPBERT-Triage) for processing multilabel neurological medical texts. We uses text-classification methods to process, analyze, train, and verify the chief complaint and diagnoses of 299,827 neurosurgical medical records of T hospital as of May 2020. The complaint is the content of the patient’s own symptoms or signs, which is a medical term. The real medical record dataset of the hospital is quite different from the public text classification dataset, which brings difficulties to the classification of this study.

(a)T hospital is specialized in neurology, and there is no research on the triage of subspecialties.(b)The chief complaint of medical records in the traditional triage method is refined by the triage personnel or doctors, while the chief complaint of the input TPBERT-Triage system in this research scenario is mainly proposed by nonmedical patients, lacking unified and standard terminology.(c)There is a high degree of similarity between the medical records of different diagnoses, resulting in the existence of a large number of "class overlapping" data; that is, different categories contain similar or identical samples.(d)There is data imbalance in the diagnosis, and the number of each diagnosis is significantly different.

In view of these problems, it is urgent that we design a suitable scheme for neurological diseases and apply natural language processing technology to the triage of subspecialties. The purpose of this study is to design a reliable intelligent triage system for subspecialty triage. The system needs to overcome the problem of imbalance and class overlapping of datasets. The TPBERT-Triage system aims to provide patients with intelligent triage services on the mobile terminal, create a triage model in line with the actual situation of the hospital, help patients to conduct self-service triage, and make a preliminary judgment on the disease, especially for specialist diseases, so as to reduce the probability of hanging the wrong department.

The main contributions of this study are as follows.

•This study starts with neurological diseases, and for the first time establishes an intelligent triage model for neurological diseases based on deep learning. It innovates the theory and mode of outpatient triage management in neurology. Without the two steps of refining the chief complaint by the triage staff and triaging the chief complaint by the triage staff, the system created in this study can be predicted based on the patient’s self-reported chief complaint. The method enhances the precision and effectiveness of neurology subspecialty triage.•According to the characteristics of neurological diseases, the degree of urgency and the hospital system, an improved label powerset (LP) method is designed. According to the diagnostic priority, the diagnostic combination is relabeled, so that the multilabel problem is transformed into a single label problem, which achieves better classification results.•In this study, a cost-sensitive learning method is used to solve the problem of data imbalance. A loss function combining focal loss and focal Tversky loss is introduced, so that the model training can pay more attention to the minority class, thereby improving the performance of text classification.

The remainder of this paper is organized as follows: related works are presented in Section 2, the framework of the system is presented in Section 3, and lastly, the experimental process and results are presented in Section 4.

## 2. Related Works

### 2.1. Text Classification

Text classification is a very classic problem in the field of natural language processing. With the development of deep learning, neural network models have been gradually introduced into natural language processing to deal with various downstream tasks and become the mainstream method. When applying deep learning to solve large-scale text-classification problems, it is most important to solve text representation. In recent years, the BERT model [11] has been widely used to encode the relationship between the markers that make up a sentence through a multilayer attention mechanism. In the text classification task, it has been proven that the BERT classifier is superior to the traditional classifier.

The BERT consists of 340 M parameters, trained with 3.3 billion words, and is the most advanced embedding model in the past three years. The trend of using larger models and more training data continues. OpenAI’s latest GPT-3 model [12] contains 170 billion parameters and has excellent performance in tasks such as translation, question answering, and text filling. Google’s GShard [13] contains 600 billion parameters. The Pangu NLP model released by Huawei is a 110-billion-parameter Chinese pretraining model trained with 40 TB data.

At present, there have been studies on training BERT models with biomedical texts as corpora and performing downstream tasks of natural language processing [14,15]. Based on the convenience and mobility of BERT, BERT has more open-source pretraining models. This study chooses BERT as the main algorithm model.

Most public datasets are ideal and clean, and there are few class overlapping cases [16,17]. The existing text classification methods do not design for this phenomenon. The class overlapping situation has a great impact on the classification performance, which brings difficulties to the real dataset of hospitals classified in this study.

In this study, medical records have a chief complaint, but the diagnosis can be more than one. Therefore, predictive diagnosis based on complaints is a multilabel text classification problem, which means that a data may have one or more labels.

The mainstream multilabel classification methods can be divided into problem-based methods and algorithm-based methods according to the classification of learning methods. The most intuitive is the binary relevance method, which transforms the multilabel classification problem into multiple binary classification problems. If a binary classification model is established for each diagnosis, the number of models will be large. On the landing scene, it is undoubtedly a waste of time resources to classify each text on multiple models. Yang et al. [18] proposed the multilabel classification task as a sequence-generation problem, and then considered the correlation between labels. The decoding part of the sequence-generation model is modified, which not only considers the correlation between tags, but also automatically obtains the key information of the input text (attention mechanism). Experiments show that the proposed method is very effective, the index is much higher than baseline, and it also has a very good effect on relationship representation.

The LP method is also a high-order strategy. This method transforms the multilabel problem into a single-label multiclassification problem. Each different label combination is considered as a different class. It is a high-order strategy that considers the correlation between multiple labels. The disadvantage is that when the label space is large, it is easy to cause overfitting. In this study, the LP method is also selected as a problem transformation method. By using this method to process the T hospital dataset, more than 1500 diagnostic combinations will be generated. Even if only the diagnosis with more than 100 occurrences is considered, there are 366 combinations. The multicategory nature is not the main reason for affecting classification, and too many categories and the class overlapping problem described above will cause great difficulties in classification.

However, there is no multilabel classification method to solve the problem of too many categories and class overlapping. Therefore, a more reasonable diagnostic combination integration scheme is needed.

### 2.2. Cost Sensitivity

In the actual classification task, the problem of data imbalance [19] is often faced; that is, in a dataset, the number of instances of each class is significantly different. Cost sensitivity is the main method by which to solve the problem of unbalanced data from the algorithm level. It uses different penalties for different classification errors when solving classification problems. The core element is the cost matrix [20].

The loss function is used to estimate the degree of inconsistency between the predicted value and the real value. The smaller the value, the better the robustness of the model. The loss function is the core part of the empirical risk function and an important part of the structural risk function.

The standard classification loss function such as cross-entropy loss (CELoss) is defined as
(1)$$L_{\mathrm{CE}}\left(\hat{y},y\right)=-\sum_{i}^{n}y_{i}\log\left({\hat{y}}_{i}\right).$$

For a specific example $y_{i}$$\left(x_{i}\right)$, regardless of the prediction results, the final error will remain unchanged [21]. ${\hat{y}}_{i}$ is the model’s estimated probability for the class with $y_{i}=1$.

Wang et al. [22] proposed a loss function with better performance in the case of unbalanced datasets, and used this loss function to calculate the errors of correct classification and misclassification, respectively. Cheng et al. [23] innovatively used the triple loss function to improve the recognition performance of convolutional neural networks for multichannel face images.

The deep neural network obtains the appropriate network weights and bias values by continuously optimizing the objective function to minimize the total loss value. By improving the loss function, the network can be cost sensitive. The common way is to add the cost parameter ω, and set different values of ω for different classification methods.

Lin et al. [24] proposed focal loss, which is defined as
(2)$$L_{\mathrm{Focal}}\left(\hat{y},y\right)=-\sum_{i}^{n}y_{i}\alpha {\left(1-{\hat{y}}_{i}\right)}^{\gamma }\log\left({\hat{y}}_{i}\right).$$

In the formula, α is the weighting factor, and γ is the focusing parameter. The focal loss function has been widely used in multiclassification tasks. It is based on the standard cross-entropy loss function. The goal is to overcome the problems of uneven sample types and uneven difficulty of sample classification. This function can make the model focus more on the samples that are difficult to classify by reducing the weight of the samples that are easy to classify. Compared with the error in correct classification, focal loss has a greater error penalty in error classification.

Although focal loss has greatly promoted the development of single-stage detection and anchor-free detection, it is also problematic. First of all, it is the core issue: it is certainly inappropriate for the model to pay too much attention to the samples that are particularly difficult to be separated. Because there are outliers in the samples, the model may have converged, but the existence of these outliers still makes it difficult to judge the existence of these outliers. Normal training, even if such a sample model is fitted, is unreasonable. Secondly, the two hyperparameters of focal loss need to be artificially designed and jointly tuned, because they affect each other. In order to solve these two problems, Li et al. proposed the gradient harmonizing mechanism (GHM) [25] to deal with this phenomenon. Based on this, the GHM classification loss (GHM-C) and GHM regression loss (GHM-R) can be easily embedded into the classification loss, such as cross-entropy, and the regression loss, such as Smooth L1. These two losses are used for the classification of anchors and the correction of bounding boxes, respectively. Experiments show that without laboriously challenging the hyperparameters, the GHM-C and GHM-R can be easily embedded into the classification loss, such as cross-entropy, and the regression loss, such as Smooth L1. GHM-C and GHM-R can bring substantial improvements to the single-stage detector and outperform the SOTA method by using focal loss and Smooth L1.

Huang et al. [26] used multiple convolutional neural networks in parallel to deal with unbalanced data problems. At the same time, they redefined the distance between classes from a geometric perspective and proposed a new error-calculation method. Khan et al. [27] has made outstanding contributions in improving the loss function. Different from other methods that only add cost factors, their proposed CoSen convolutional neural network optimizes both network parameters and cost parameters. Experiments are carried out on a variety of classical loss functions (MSE loss, SVM loss, CE loss) to improve the classification accuracy of convolutional neural networks.

Yeung et al. [28] evaluated which composite loss function can effectively deal with class-imbalance problems.It can be seen from these related works that different loss functions should be selected for training according to different situations of downstream tasks and the degree of sample imbalance. This paper tests the performance of text classification under different loss functions.

### 2.3. Intelligent Triage

In recent years, computer technology has been widely applied in the medical field [29,30] and has played an increasingly significant role [31,32]. At present, natural language processing technology has been used in triage and related directions.

Garla [33] developed a clinical text classification system based on machine learning, which uses the classification structure of the unified medical language system to improve the feature ranking. The author also proposed a context semantic similarity evaluation method to project clinical texts into a feature space. To improve the classification accuracy. Mujtaba et al. [34] classify clinical texts based on the SML method, and apply feature engineering to extract the most discriminative features from clinical reports to form numerical feature vectors. This feature vector is used as the input of the learning algorithm to construct and verify the classification model. The above research mainly focuses on rule-based or feature engineering, which is a traditional machine learning method.

After the emergence of the neural network, it developed at an alarming rate. Deep learning has been applied to various scenarios in real life, including the medical industry [35]. Its advantage is that it can achieve many practical applications that machine learning cannot meet. Other research has seen applications, such as digital twinning technology and blockchain technology in medical condition monitoring and virtual diagnosis and treatment [36,37]. In the study of Yao et al. [38], a method of combining rule-based features and knowledge-guided deep learning models for effective disease classification was proposed. The key steps of this method include identifying trigger phrases, using trigger phrases to predict classes in few examples, and using word embedding and unified medical language system (UMLS) entity embedding to train convolutional neural networks. Rios et al. [39] used convolutional neural networks to construct a binary text classifier. On a set of MeSH terms that are difficult to classify, the F1 value of the model is improved by more than 3% compared to the best prior result on the public dataset. It is feasible to use convolutional neural networks to represent the semantics of clinical texts [40], and semantic classification can be performed at the sentence level. Compared with shallow learning methods, multilayer convolutional deep networks can generate more optimal features to represent the semantics of the analyzed sentences in the training phase [41].

For intelligent triage, artificial intelligence and other technologies are used in the direction of triage. Related research first appeared in the emergency department, because emergency department crowding has become a public health problem around the world [42]. Due to the heavy workload of the hospital and the lack of inpatient beds, patients wait too long between the emergency department to the inpatient ward, which directly affects the mortality of patients [43]. In this regard, some studies use logistics regression. Using the data collected by the hospital during the traditional triage to establish a prediction model, which can predict whether the patient needs to be hospitalized at the time of triage, can help to determine the patient’s hospitalization plan and resource allocation [44].

With the development of natural language processing technology, there are many studies on the design of triage system using unstructured data [7,45,46]. The unstructured data used is mainly the chief complaint, because the chief complaint represents the reason for the patient’s visit. The current triage system solves the triage for general practitioners.

The current triage system deals with triage for the general practice. However, there are differences in datasets between general practice triage and subspecialty triage. In the triage of general practice, the text descriptions of medical records in different departments differ greatly, and the classifier is easier to fit. For the subspecialty triage, the chief complaint text has similar characteristics, The complexity of medical records is reflected in the neurosurgical records processed in this study. There is no specific study to classify subspecialty cases in more detail.

## 3. TPBERT-Triage System Design Principle

This section introduces the system structure and design ideas of the intelligent triage system. In order to improve the efficiency and accuracy of triage, and effectively reduce the rate of withdrawal and number change, this study established a set of intelligent triage system: TPBERT-Triage, which can use medical record text and its diagnostic combination training to predict diseases.

The main process of the TPBERT-Triage system includes transforming the medical record data into the target dataset format that the system can read; the system uses the target dataset to fine tune the pretraining model in advance, generates a prediction model, and loads it into memory. When the back end of the system receives the patient ’s complaint from the client, the system is based on the BERT model, and uses the text-classification method to predict the most likely diagnosis or diagnostic combination according to the chief complaint, returns it to the patient, and recommends the specific department to the patient.

### 3.1. The System Design

The TPBERT-Triage system developed in this study is shown in the Figure 1, which is mainly composed of two modules: fine tune and predict.

The fine-tune module is the core module of the scene, which completes the training process of the pretraining model used in the TPBERT-Triage system prediction in advance. First, data desensitization and resampling are performed on medical record data to overcome the imbalance of medical records. Secondly, the medical record data is preprocessed and transformed into the data format required for text classification. The text corresponds to the chief complaint of the medical record, and the category corresponds to the diagnostic combination of the medical record. Furthermore, the preprocessed data set is transformed into TFRecord format and introduced into the BERT pretraining model for training and evaluation. The final pretrained model can be loaded into the system for prediction. The predict module uses the pretrained model trained by the fine-tuned process to predict. When the patient enters the chief complaint in the client, the chief complaint is preprocessed and passed into the BERT model. Finally, the TPBERT-Triage system returns the category and recommended department to the patient.

Based on big data, artificial intelligence, natural language processing and other related technologies, the TPBERT-Triage system provides patients with services such as triage of corresponding diseases, recommendation of departments, and even recommendation of corresponding subspecialists based on the symptoms input by patients, with the goal of realizing precise medical treatment for outpatients, innovating hospital service mode and further providing effective experience and basis for graded treatment.

The TPBERT-Triage system uses Huawei’s self-developed domestic AI framework Mindspore for training. Through the second-order optimization method, it accelerates the decline process of the loss function loss value and reduces the deep learning training time [47].

### 3.2. Design Ideas and Methods

The dataset of this study is based on the medical records of neurosurgery in T hospital. The main fields of medical records are gender, age, chief complaint, current medical history, past medical history, allergic history, physical examination, diagnosis, symptoms, etc. The landing scene of TPBERT-Triage system is that after the client describes his symptoms, the system matches the appropriate department for the patient according to the symptoms.

For the landing scenario of the TPBERT-Triage system, this study selects the chief complaint as the classification text because the chief complaint is relatively short. This refers to the triage system designed by Wang et al [46]. In view of the fact that patients are often unable to examine themselves, there are no professional instruments such as CT and MRI around them, and it is impossible to obtain more detailed symptoms and specific values of various signs, such as blood pressure and pupil, this study does not use physical examination as a classification text; even if the examination has more information and elements, it is difficult for patients to complete self-test and accurately describe their symptoms.

The triage system of the hospital is not static. The same diagnosis will be assigned to different departments under different situations and systems. In response to this problem, the TPBERT-Triage system is designed to use diagnosis as a label rather than a department as a label. Because the current medical record dataset is not labeled based on the rules of named entity recognition, manual labeling will undoubtedly produce huge inestimable costs, and manual labeling requires medical professionals to complete, this study finally adopts the text-classification method; it only needs to use the diagnosis of medical records as a label, eliminating the process of data labeling, making data processing simpler and more time efficient, and easier to implement in engineering. In summary, when training and predicting the model, the author chooses the chief complaint and diagnosis as the text and label for the text classification task.

The fine-tune module and the forecast module make up the TPBERT-Triage system. The corresponding role is assigned to each module. The system’s collaborative development, system debugging, and subsequent expansion and upgrading are all made possible by modular development. The secret to the fine-tune module is how to specify the classified text and category so that the trained model can forecast the appropriate category of the text based on various texts. The BERT model extracts the semantics from the chief complaint in this research so that the trained model can catch more complex word meanings from the chief complaint. The diagnosis combination of medical records is integrated based on the diagnosis priority scheme. The following elaborates on these aspects.

#### 3.2.1. Triage Priority Design

Aiming at the problem that the medical record text corresponds to multiple diagnoses, the study initially used the LP method for category labeling; that is, each different diagnostic combination is treated as a category, thereby transforming the multilabel problem into a single-label problem. When dealing with real datasets in reality, the data is not as ideal as many public text classification datasets [48,49]. First, there are more diagnostic combinations. Even if only the diagnosis with more than 100 occurrences is considered, there are 366 diagnostic combinations. Secondly, there is the problem of class overlapping. For example, the medical records with the same chief complaint of “intermittent headache for 1 h, progressive aggravation” have a total of 34 diagnostic combinations, of which 11 include intracranial space-occupying lesions. The diagnostic combination also includes 31 diagnoses such as epilepsy and trigeminal neuralgia. The reason why the chief complaint is the same but the diagnosis is different is that the doctor also needs to analyze the patient’s physical examination, past medical history, etc., so as to obtain the diagnosis, and the elements and information contained in the chief complaint are not as rich as the physical examination and past medical history, so the chief complaint and diagnosis are not a one-to-one relationship. This situation is called the class overlapping problem in the classification problem; that is, some samples from different categories have very similar features or even the same situation. The training of data sets with more class overlapping problems will make the loss function value unable to converge, and will also affect the accuracy of classification and other indicators. Therefore, this study integrates these diagnoses.

Aiming at the problem of too many diagnostic combinations with class overlapping and data imbalance, this study consulted the medical staff of T hospital, designed a diagnostic priority definition scheme based on professional knowledge, hospital system, and the characteristics of neurological diseases, and integrated and summarized the diagnostic combinations. The problem transformation method based on this scheme is called triage priority (TP). In the dataset of this study, there are 28 diagnoses with more than 100 occurrences. In this study, these 28 diagnoses were integrated into 10 categories according to the characteristics of neurological diseases, severity, and hospital system. As shown in Table 1, head trauma, scalp laceration, etc. belong to the category of trauma, and the injury of such patients needs immediate treatment, so the priority is the highest; the priority of cerebral hemorrhage is second, and the priority of epilepsy is third.

When the diagnosis of head trauma and subarachnoid hemorrhage occurs at the same time, the case is classified into trauma category, because head trauma belongs to trauma category, and subarachnoid hemorrhage belongs to cerebral hemorrhage category. By defining diagnostic priorities, the diagnostic portfolio was consolidated, reducing the number of categories and reducing class overlapping data. If there are three diagnostic combinations with the same chief complaint, then after integrating the summary diagnostic combinations, only two categories may have the same chief complaint, and even these three diagnostic combinations are integrated into one class, which overcomes the problem of class overlapping.

There may be a correlation between tags [50]. For the similar complaints in different diagnoses, after the integration of the diagnostic priority scheme, the similar complaints will be basically classified into the same category; for similar complaints in different categories, the diagnosis priority scheme considers the severity of the disease, does not fully consider the correlation between labels, has limitations, but also conforms to the landing scene and hospital situation of the system.

In summary, on the one hand, the diagnostic priority definition scheme effectively reduces the number of classification categories and improves the classification accuracy. On the other hand, it takes into account the mitigation and severity of the disease from the actual needs, which helps to prioritize the patients with more conditions to higher priority categories.

#### 3.2.2. Problem Transformation

The types of neurological diseases are complex and diverse, which are manifested as follows.

(1)The same chief complaint but different diagnosis. Clinically, many diseases often start with the same symptom, which is easy to cause confusion and cause wrong triage. For example, the chief complaint is “being hit by a car, injuring the head for more than 3 h”, but there are two different diagnoses: multiple brain contusion, subarachnoid hemorrhage, skull fracture, intracranial pneumatosis, scalp hematoma and head trauma. The main complaint is “intracranial space occupying requires infusion”, and the diagnosis has “intracranial space occupying lesions”, but some diagnoses also include brain edema.(2)The disease involves multiple diagnoses. According to the neurosurgery medical records of T hospital, each medical record corresponds to 1.2 diagnoses on average.

This research processes multilabel medical records using a variety of problem-transformation techniques and compares them through experiments based on the characteristics of neurological diseases. The methods used for ablation experiments include BR, classifier chain (CC [51]), etc.

Two methods for problem transformation were developed in this research.

The first method is dictionary mapping (DM), which is independent of the dataset itself. The data format before the DM method conversion is shown in Table 2. Since the dataset of this study is multilabel, a text may correspond to a single or multiple categories. The DM method attempts to transform the multilabel problem into a multiclassification problem. The idea is that when a text corresponds to multiple categories, the collection of these categories contains the text. As shown in the data format of Table 3, DM method is simple to process, and the data obtained will not have too many categories, and the integration of categories is relatively simple. This method provides a high classification accuracy in the first stage of this study; the shortcomings of the DM method are also obvious. This method will directly lead to the class overlapping problem. Even if each complaint in the dataset corresponds to an average of 1.27 labels, the data set itself contains the class overlapping problem. This method does not alleviate this problem but generates more class overlapping data, which aggravates the impact of class overlapping.

The second method is the triage priority (TP) method, which itself is related to the dataset. The top 10 categories of sample quantity after LP transformation are shown in Table 4; there is an overlapping between the categories. In this study, an improved LP method is proposed, the TP method. The TP method summarizes the categories contained in each other according to the diagnostic priority definition scheme in Section 3.2.1, such as categories 1, 3, 6, 8, and 10 in the Table 1 are classified as trauma classes, thereby reducing the impact of the class overlapping problem. The LP method and the TP method perform problem transformation as shown in the Table 5. Different label combinations are regarded as different LP labels, while the TP label is defined according to the priority of the label, in all labels of the text. Take the highest priority as the only label.

#### 3.2.3. Linguistic Representations

BERT is an algorithm model for language representation in this study, which is mainly used to learn the semantics of the chief complaint text. BERT selects two tasks simpler than the language model for pretraining: cloze and sentence pair prediction [11]. In the cloze task, some words need to be masked, so the pretrained model is flawed because these words cannot be masked in other tasks. BERT solves this problem in a random way: 80% with mask, 10% with other words, 10% with the original word. This model has the ability to migrate.

Under the unique training method, BERT can better predict based on context and have a better understanding of context relationship. The BERT pretraining model used in this study is a Chinese pretraining model based on Mindspore 1.3.0 trained with the Ascend310 processor. In the training process, a hybrid precision training method is used to accelerate the training process of deep neural networks by mixing single-precision and semiprecision data formats, while maintaining the network accuracy that can be achieved by single-precision training [52]. Hybrid precision training can accelerate the calculation process, reduce memory usage and access, and enable larger models or batch sizes to be trained on specific hardware. After pretraining, it can be fine tuned according to downstream tasks. The downstream task of this study is text classification. The overall process is two parts: after the embedding representation of the original text is obtained by BERT, the embedding is put into the fully connected layer for classification. In order to improve the performance of the model, we often add a dropout layer before the linear layer, which can reduce the possibility of overfitting of the network [53] and enhance the independence of neurons.

Guiding patients to describe their symptoms and generate complaints is one of the primary tasks of triage. The chief complaint helps to determine the main reason for patients seeking treatment. Therefore, the need to accurately capture and specifically respond to patients’ complaints fundamentally affects the way hospitals record complaints. The most common way of recording the chief complaint is transcribed into the medical record by the triage personnel according to the description of the patient. It can be seen that the chief complaint is narrated by the professional personnel according to their own professional knowledge, which is professional, concise, and relatively consistent. The term “professional” means that the words of the chief complaint are accurate and professional, standardized, and rigorous, and try to use medical terms and avoid colloquialism; The term “conciseness” means that the chief complaint reflects the most important symptoms or needs of the patient with the least number of words, and the words are refined.

Relative consistency means that different professionals have similar complaints for similar patients, even if the patient’s description is completely different. Aiming at the problem that patients cannot extract their own symptoms due to lack of medical expertise, this study uses the BERT model for language representation, because BERT learns special surface, syntactic and semantic feature representations [54] at each layer. The model captures the high-level semantics of words from the patient’s description, understands the connections of different words in different sentences, and can more accurately capture the similarity of words [15,55].

#### 3.2.4. Loss Function

Many NLP tasks are faced with data imbalance. The most commonly used cross-entropy loss function is actually a loss function for accuracy. During training, each training instance contributes equally to the objective function, while the F1 score pays more attention to positive examples during testing.

The Tversky loss [56] is found by combining two adjustable parameters when α and β are assigned to false positives and false negatives, respectively, to optimize the output imbalance. Focal tversky loss(FTL) [57] adapts the Tversky loss by applying a focal parameter. The formula is as follows,
(3)$$L_{FT}=\sum_{c=1}^{C}{\left(1-\frac{\sum_{i=1}^{L}p_{0i}g_{0i}}{\sum_{i=1}^{L}p_{0i}g_{0i}+\alpha \sum_{i=1}^{L}p_{0i}g_{1i}+\beta \sum_{i=1}^{L}p_{1i}g_{0i}}\right)}^{\frac{1}{\gamma }},$$
where $p_{0i}$ is the probability of the *i*th belonging to the foreground class, and, $p_{1i}$ is the probability of the *i*th belonging to background class. $g_{0i}$ is 1 for foreground and 0 for background and conversely $g_{1i}$ takes values of 1 for background and 0 for foreground. γ<1 increases the degree of focusing on harder examples. The FTL simplifies to the Tversky loss when γ=1.

In practice, if a sample is misclassified with a high Tversky index, the FTL is unaffected. However, if the Tversky index is small and the sample is misclassified, the FTL will decrease significantly. When γ>1, the loss function focuses more on less accurate predictions that have been misclassified. Most experiments start with $\gamma =\frac{4}{3}$. Abragam et al. [57] hypothesize that using a higher α in the generalized loss function will improve model convergence by shifting the focus to minimize FN predictions, and train all models with α=0.7 and β=0.3.

Due to the different imbalance degrees of different datasets, researchers have not reached an agreement on which one to choose between the basic loss function and the composite loss function. This paper uses a loss function that combines focal loss and FTL, including adjustable parameters to deal with output imbalance. The formula is as follows,
(4)$$L_{\mathrm{composite}}=\lambda L_{Focal}+\left(1-\lambda \right)L_{FT},$$
where λ∈[0,1] determines the relative weighting of the two component loss functions. In this paper, after adjusting parameters, λ was finally selected as 0.7.

## 4. Experiments

This study compares the effects of various problem transformation methods and cost-sensitive methods on text classification performance. The BERT-base model is used, and the training data is the chief complaint and diagnosis of neurosurgery medical records in T hospital.

### 4.1. The Experimental Process

#### 4.1.1. The Evaluation Index

The most commonly used evaluation indicators for classification are accuracy, precison, recall, F1 score, etc. According to the combination of sample real results and model prediction results, it can be divided into TP, FP, FN, and TN. In this study, the downstream task is a multiclassification problem, so for each category, there is a TP, FP, FN, and TN value. For each category, its accuracy, precision, and recall can be calculated. When the model predicts the test set, the prediction is a probability list for each label. We choose the label with the highest probability. If the label is real, the prediction is correct. Otherwise, it is wrong. In this study, accuracy, macro-average indicators (precision-macro, recall-macro and F1-macro) and weighted indicators (precision-weighted, recall-weighted and F1-weighted) were used as evaluation indicators. This selection was based on the fact that $F1_{\mathrm{micro}}$ could almost only reflect the situation of categories with large sample sizes when the data were extremely unbalanced. The formula was shown as (Equation 5)–(Equation 11); at the same time, the confusion matrix is selected as a tool for evaluating classification.
(5)$$\mathrm{Accuracy}=\frac{\sum_{i=1}^{L}\left(TP_{i}+TN_{i}\right)}{\sum_{i=1}^{L}\left(TP_{i}+TN_{i}+FP_{i}+FN_{i}\right)}$$
(6)$${\mathrm{Precision}}_{\mathrm{macro}}=\frac{1}{L}\sum_{i=1}^{L}{\mathrm{Precision}}_{i}$$
(7)$${\mathrm{Recall}}_{\mathrm{macro}}=\frac{1}{L}\sum_{i=1}^{L}{\mathrm{Recall}}_{i}$$
(8)$$F1_{\mathrm{macro}}=2\cdot \frac{{\mathrm{Precision}}_{macro}\cdot {\mathrm{Recall}}_{macro}}{{\mathrm{Precision}}_{macro}+{\mathrm{Recall}}_{macro}}$$
(9)$${\mathrm{Precision}}_{\mathrm{weighted}}=\frac{\sum_{i=1}^{L}\left({\mathrm{Precision}}_{i}\times w_{i}\right)}{\left|L\right|}$$
(10)$${\mathrm{Recall}}_{\mathrm{weighted}}=\frac{\sum_{i=1}^{L}\left({\mathrm{Recall}}_{i}\times w_{i}\right)}{\left|L\right|}$$
(11)$$F1_{\mathrm{weighted}}=2\cdot \frac{{\mathrm{Precision}}_{\mathrm{weighted}}\cdot {\mathrm{Recall}}_{\mathrm{weighted}}}{{\mathrm{Precision}}_{\mathrm{weighted}}+{\mathrm{Recall}}_{\mathrm{weighted}}}$$
where $TP_{i}$ denotes the TP value of the *i*th category, and similarly, $TN_{i}$, $FP_{i}$, $FN_{i}$, ${\mathrm{precision}}_{i}$ and ${\mathrm{recall}}_{i}$ with subscripts represent TN, FP, FN, precision and recall of the *i*th category. $\omega_{i}$ represents the percentage of the sample size of the *i*th category to the overall.

#### 4.1.2. Data Preprocessing

This study analyzed and trained the neurosurgical medical records of T hospital as of May 2020, including 299,827 medical records. The electronic medical records are filled in by the attending physician, and the background automatically forms a structured electronic medical record. The sample medical record is shown in the Table 6. One chief complaint may correspond to multiple diagnoses. According to statistics, each chief complaint corresponds to 1.27 diagnoses or 1.27 labels on average. Because the data used in this study have a long time span, and the development of electronic medical records is short, structured electronic medical records often do not have high quality, and more fields are empty or noisy. It is necessary to perform a series of operations on the data, such as data desensitization, data cleaning, diagnostic screening, label integration, and label sorting, so that the data meets the format requirements of the model training data.

Data desensitization is data deprivacy. In this study, the training of the model only uses the main complaint and diagnosis field of the medical record, and does not use the name, ID, and other fields. In addition, it is not possible to reason the medical record itself from the final prediction model, so data security is guaranteed to a certain extent. Data cleaning is to process the medical records with large noise in the text. For example, there are various obvious wrong punctuation marks in many complaints, which have no positive effect on sentence classification and need to be deleted. Finally, 25,420 medical records with both complaints and diagnosis are obtained. Diagnostic screening is used to remove the diagnosis that appears very few times in the diagnosis and has little connection with the neurology department. For example, “depression” has only appeared once, and “seasonal allergic rhinitis” has only appeared once. Of the 511 diagnoses that have appeared, only 28 have occurred more than 100 times.

#### 4.1.3. Experimental Environment

As shown in the Table 7, this study uses the deep learning framework Mindspore to encode the model and deploys it in the Ubuntu 18.04 system. The GPU uses Nvidia’s RTX3080 to accelerate the experiment. In order to more objectively evaluate the classification results of different methods, this study uses the same parameters to train the model, and the relevant parameters are shown in Table 8. The selection of hyperparameters in this paper refers to the research of Sun et al. [55]. This study uses the Lamb optimizer to optimize the model, and the initial learning rate is 0.00002; the Dropout method [53] is used to prevent overfitting, and the Dropout of the word embedding layer is set to 0.1.

### 4.2. Results and Analysis

For example, the confusion matrix shown in Figure 2 and Figure 3 allows analysis of the classification performance of LP methods before and after the integration of the diagnostic prioritization scheme. Most of the samples in five categories are classified as category 1 because the first category has the largest proportion of samples and has a larger weight in the training. Among them, category 6, category 8, and category 10 were incorrectly predicted, and all of them were predicted as category 1. Upon analysis, head trauma was the most numerous in category 1, while category 6 was head trauma + scalp lacerations, and categories 8 and 10 both contained head trauma, so most of the chief complaints in these categories were similar to category 1. During training, these categories were not weighted as heavily as the first category, and the sample was easily assigned to the first category. After the diagnostic priority integration scheme, as shown in Figure 4, only two categories were predicted incorrectly in all cases.

According to the diagnostic priority definition scheme, traumatic brain injury syndrome and head trauma belong to the highest priority trauma category, and the two are classified into one category. Under the TP method, the category integration is carried out according to the diagnostic priority definition scheme. The text classification accuracy and recall rate of the trauma class are 98.45% and 93.21%, respectively, and 98.85% and 76.99% of the head trauma class in the LP method. The scheme is effective in improving the accuracy and recall rate.

We conducted a study to observe the convergence of loss values during model training by using various problem-transformation methods. The loss value is a direct indicator of the model’s fit on the training set. Our results, shown in Figure 5 and Figure 6, demonstrate that both the DM and LP methods trained without integration have large loss function values and do not show any signs of convergence.

As shown in the Table 9, the overlapping text of the original data accounts for 9.66%, and the proportion is 14.75% after DM processing, which is increased by 52.69%. For example, for text 1, text 1 appears in both labels. When the model trains text 1 and label 1, the hidden layer is changed by backpropagation. When the model trains text 1 and label 2, the hidden layer of the model changes again, so the loss function value of model training will always oscillate.

After the integration of the diagnostic integration scheme, the proportion of class overlapping text of the integration-DM method and the integration-LP method decreased by 47.25% and 19.24%, respectively, the loss function value decreased significantly, and the oscillation amplitude decreased.

For the TP method, the class overlapping text proportion of the data is the lowest among several problem-transformation methods. Since the data set itself has class overlapping text, the loss function value of the model does not always decline and is oscillating. This kind of oscillation is unavoidable for the real hospital data set of this study.

In this study, the macro-average index (Macro) is used to measure the global classification performance; at the same time, because of the large difference in the number of samples in the dataset classified in this study, in order to solve the problem that the previous indicators cannot measure the sample equilibrium, this study also calculated the weighted index (weighted). As shown in Table 10, we have the following observations.

The TP method has the highest accuracy of all problem-transformation methods, which is 87.47%. However, due to its large data imbalance index, it does not improve the macro-average index compared with the other two integrated methods. After the diagnostic integration scheme is processed, the integration-DM and integration-LP methods significantly improve the accuracy and macro-average indicators than the DM and LP methods.

The classification results after using other loss functions are shown in Table 11. From the experimental results, it can be seen that using the original CELoss unexpectedly performs better than using several other loss functions for unbalanced data. The composite loss function also only improves the accuracy by 1.03%, while the macro-average index is 28.93%, 29.75%, 26.43% lower than CELoss.

Although different loss functions do not significantly improve the classification accuracy in this study, this study shows that selecting an appropriate loss function for imbalanced datasets can help improve the classification performance of minority classes.

## 5. Conclusions

In this study, we designed a deep learning-based intelligent triage system, TPBERT-Triage. The system utilizes the BERT model and text classification methods to categorize neurosurgery medical records from T hospital. The system combines the chief complaint and diagnosis as text and category for classification purposes. BERT is capable of capturing the high-level semantics of words, thus eliminating the need for professional triage personnel to input chief complaint. Patients can input self-reports via their mobile devices. Then, according to the professional knowledge of hospital professionals, this study designed a definition scheme of diagnostic priority, including ten categories such as trauma, cerebral hemorrhage and epilepsy. The TP method was used to transform the multilabel medical record text, and 366 diagnostic combinations were integrated and summarized to reduce the number of categories and the proportion of class overlapping of data. The experimental results show that the appropriate and scientific integration scheme helps to improve the performance of the model for diagnostic prediction. The integration scheme with priority can reduce the class overlapping of text and improve the accuracy of text classification. Using TP method has better performance for evaluation index and model convergence. Compared with no integration, the use of diagnostic priority definition scheme integration improves the classification accuracy by more than 50%. In order to raise the classification accuracy to 88.38%, this research compares a number of cost-sensitive learning experiments and chooses a composite loss function. Our current triage system has a few limitations from the clinical perspective. First, the current study only addressed triage in neurology and did not yet transfer the protocol to other subspecialties. In addition, the triage only referred to the chief complaint and did not use other information, such as physical examination.

In future work, we will seek to introduce more multimodal medical history information such as images, apply multimodal learning methods, and train with information such as physical examination and medical history, which will make triage achieve better results. Finally, with neurosurgery as a breakthrough, the intelligent triage system is established and gradually extended to more subspecialties.

## Figures and Tables

**Figure 1 bioengineering-10-00420-f001:**
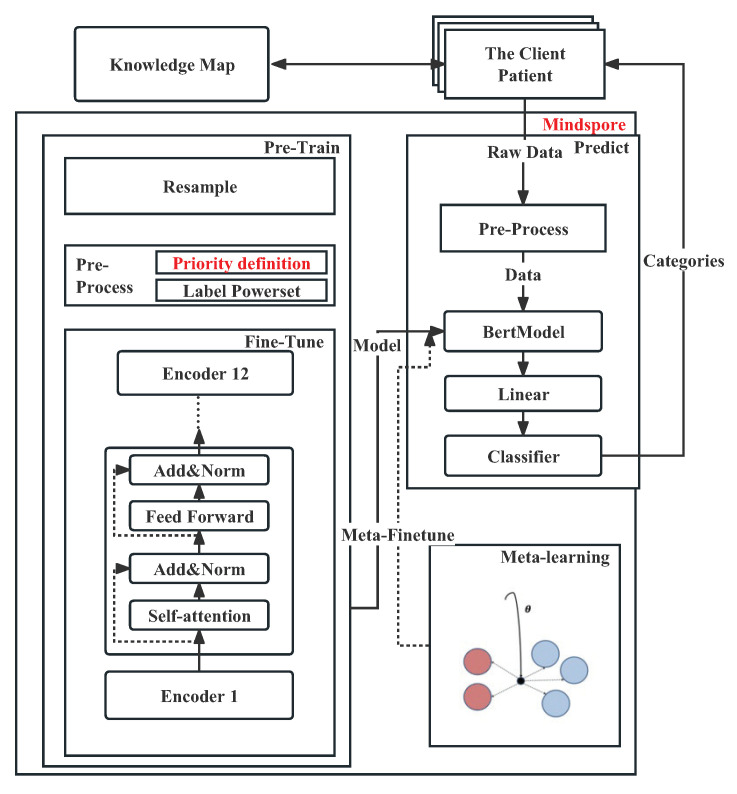
The system structure.

**Figure 2 bioengineering-10-00420-f002:**
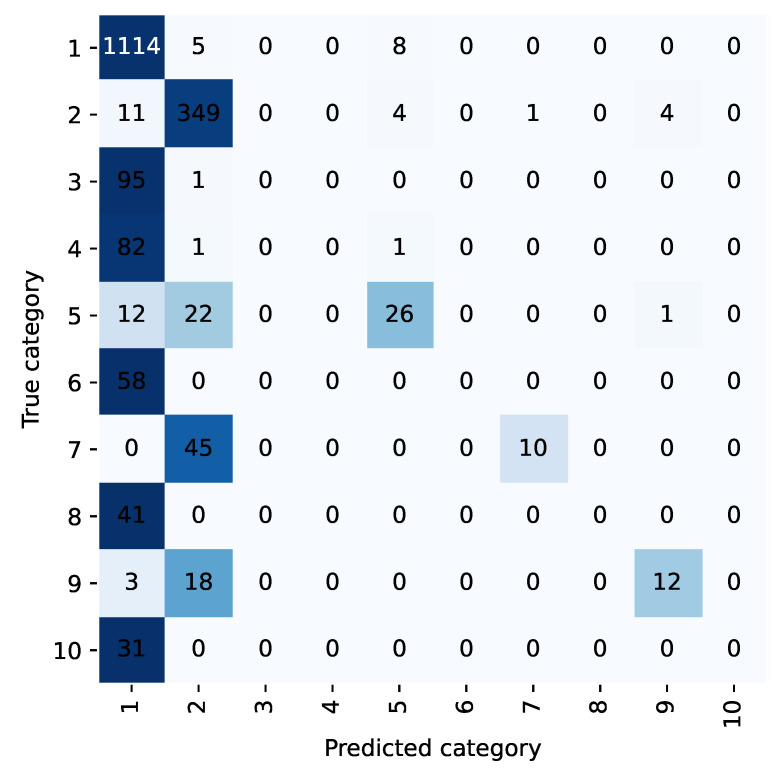
Confusion matrix of LP method.

**Figure 3 bioengineering-10-00420-f003:**
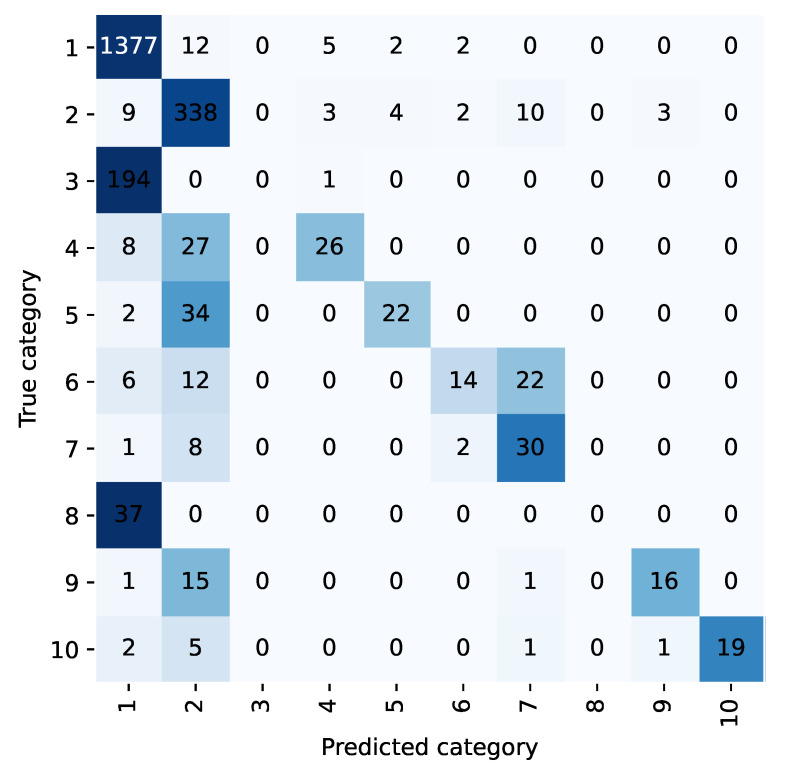
Confusion matrix of integration-LP method.

**Figure 4 bioengineering-10-00420-f004:**
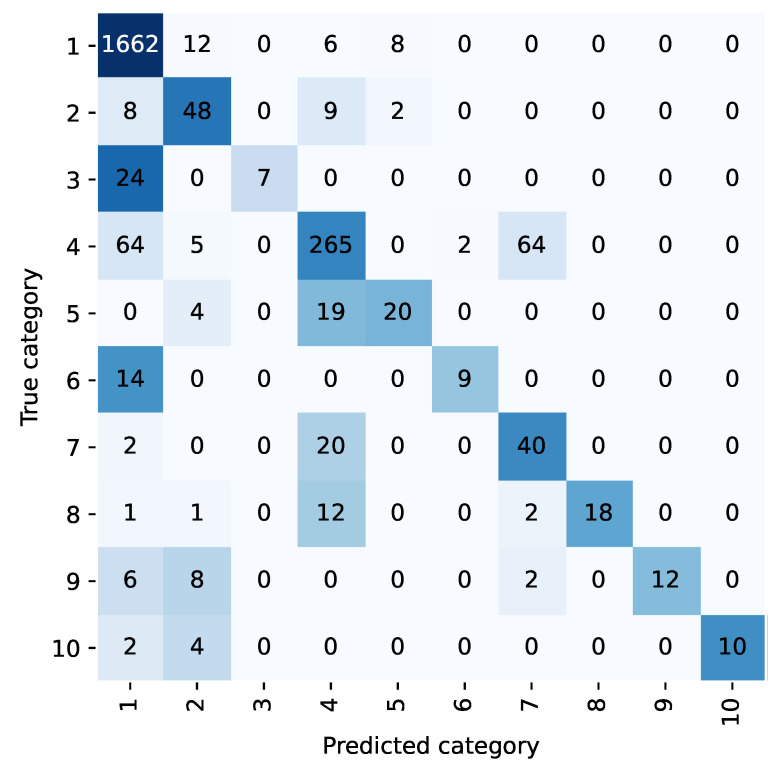
Confusion matrix of TP method.

**Figure 5 bioengineering-10-00420-f005:**
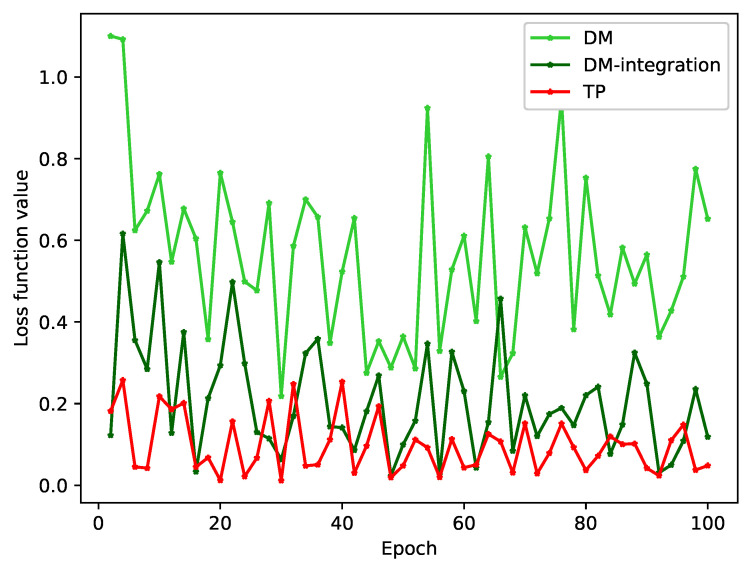
The convergence (oscillation) of the loss function values of DM, integration-DM, and TP methods.

**Figure 6 bioengineering-10-00420-f006:**
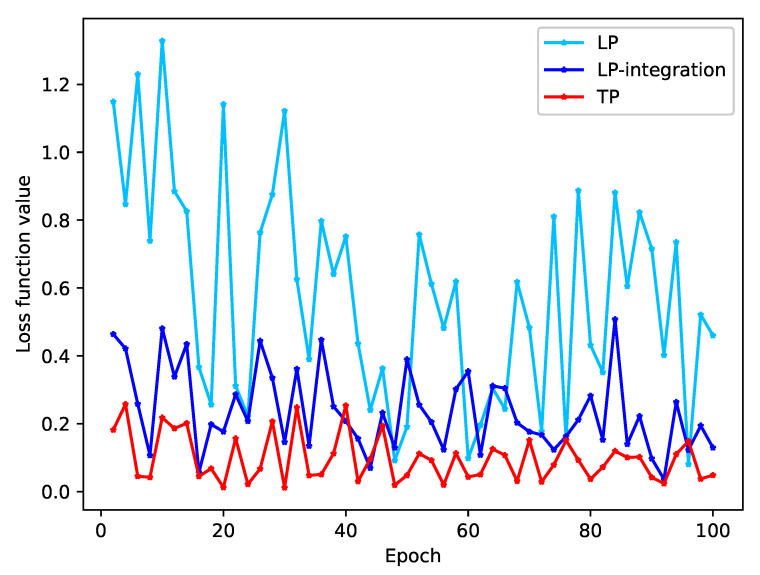
The convergence (oscillation) of the loss function values of LP, integration-LP, and TP methods.

**Table 1 bioengineering-10-00420-t001:** Triage priority scheme and number of categories.

Priority Label	Diagnosis Containing	Number of Data
1	Head trauma, Scalp laceration, Traumatic brain injury syndrome, Scalp hematoma, skull fracture, scalp contusion, Subarachnoid hemorrhage, multiple cerebral contusion, Traumatic epidural hematoma, Traumatic subdural hematoma, Traumatic cerebral hemorrhage, cerebral edema, Traumatic intracranial hemorrhage	17,206
2	Subarachnoid hemorrhage, cerebral hemorrhage	664
3	Epilepsy	195
4	Intracranial space occupying lesions	4000
5	Intracranial aneurysm, cerebrovascular malformation, Aneurysm	427
6	After the brain tumor surgery, After the pituitary tumor surgery	239
7	Chronic subdural hematoma	622
8	Hydrocephalus	336
9	Headache	285
10	Dizzy	51

**Table 2 bioengineering-10-00420-t002:** The data format before problem transformation.

Text	Label1	Label2	Label3	Label4	Label5	Label6
text1	1	0	1	0	1	0
text2	1	1	0	0	0	0
text3	1	0	0	0	0	0
text4	0	1	0	0	0	0
text5	0	0	1	0	0	0

**Table 3 bioengineering-10-00420-t003:** The data format after transformation by DM method.

Label	Text
Label1	text1
Label3	text1
Label5	text1
Label1	text2
Label2	text2
Label1	text3
Label2	text4
Label3	text5

**Table 4 bioengineering-10-00420-t004:** The first ten categories after transformation by LP method.

Label	Diagnosis of Combination	Number of Data
1	Head trauma	11,267
2	Intracranial space occupying lesions	3686
3	Scalp laceration	958
4	Posttraumatic brain injury syndrome	836
5	Chronic subdural hematoma	615
6	Head trauma+scalp laceration	576
7	Intracranial space occupying lesions +brain edema	549
8	Head trauma+scalp hematoma	410
9	Hydrocephalus	333
10	Head trauma+posttraumatic brain injury syndrome	307

**Table 5 bioengineering-10-00420-t005:** The data format after transformation by LP/TP method.

Text	Label1	Label2	Label3	Label4	Label5	Label6	LP-Label	TP-Label
text1	1	0	1	0	1	0	1	1
text2	1	1	0	0	0	0	2	1
text3	1	0	0	0	0	0	3	1
text4	0	1	0	0	0	0	4	2
text5	0	0	1	0	0	0	5	3

**Table 6 bioengineering-10-00420-t006:** One example of a medical record.

The Chief Complaint	Diagnosis 1	Diagnosis 2
Fall and hurt head for 2 h	Head trauma	Traumatic subarachnoid hemorrhage

**Table 7 bioengineering-10-00420-t007:** Description of the software and hardware environment.

	Experimental Environment	Environment Configuration
Software environment	The operating system	Ubuntu 18.04.5 LTS
	Programming language	Python3.7.5
	Programming framework	Mindspore1.5.0
Hardware environment	CPU	Intel Gold 5218 CPU@2.3 GHz
	GPU	NVIDIA GeForce RTX3080

**Table 8 bioengineering-10-00420-t008:** Model hyperparameter setting.

Hyperparameter	Hyperparameter Value
The number of nodes in each layer of neural network	768
The layer number of the Transformer	12
Multiple attention quantity	12
Hidden layer dropout rate	0.1
Initial learning rate	$2.00\times 10^{-5}$
The weights of decay	0.01

**Table 9 bioengineering-10-00420-t009:** Class overlap situation.

Processing Method	The Total Number	Number of Overlapping Texts	Proportion of Overlapping
original	23,975	2316	9.66%
DM	24,969	3683	14.75%
integration-DM	28,451	2213	7.78%
LP	19,552	1596	8.16%
integration-LP	22,746	1499	6.59%
TP	23,975	943	3.93%

The integration-DM method and integration-LP method mean DM method and LP method using the triage priority scheme, Table 1.

**Table 10 bioengineering-10-00420-t010:** Evaluation index values of various processing methods.

Processing Method	Accuracy	Precision-Macro	Recall-Macro	F1-Macro	Precision-Weighted	Recall-Weighted	F1-Weighted
BR	0.3086	0.1153	0.1636	0.1352	0.1414	0.1716	0.1550
CC	0.5620	0.4038	0.3223	0.3585	0.5417	0.5606	0.5509
DM	0.7372	0.7372	0.7372	0.7372	**0.8563**	0.8522	0.8543
integration-DM	0.8473	**0.8473**	**0.8473**	**0.8473**	0.8401	0.8467	0.8434
LP	0.7729	0.3843	0.2906	0.3309	0.6515	0.7729	0.7070
integration-LP	0.8039	0.6065	0.4361	0.5074	0.7240	0.8039	0.7619
TP	**0.8747**	0.8334	0.8005	0.8166	0.8522	**0.8747**	**0.8633**

**Table 11 bioengineering-10-00420-t011:** Evaluation index values of various loss function.

Processing Method	Accuracy	Precision-Macro	Recall-Macro	F1-Macro	Precision-Weighted	Recall-Weighted	F1-Weighted
BERT+CE	0.8747	**0.8334**	**0.8005**	**0.8166**	0.8522	0.8747	0.8561
BERT+Focal	0.8734	0.4195	0.3713	0.3939	**0.8705**	0.8734	**0.8667**
BERT+Class- Balanced Focal	0.8535	0.4867	0.5505	0.5166	0.8213	0.8535	0.8325
BERT+Tversky	0.8643	0.5552	0.5964	0.5751	0.8019	0.8643	0.8234
BERT+FTL	0.8672	0.5328	0.5115	0.5219	0.8517	0.8672	0.8524
BERT+ourwork	**0.8838**	0.5923	0.5855	0.5889	0.8481	**0.8838**	0.8576

The best values for variouslosses are reported from the hyperparameter tuning.

## Data Availability

Not applicable.

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
