# Peer review of "A Disease-Prediction Protocol Integrating Triage Priority and BERT-Based Transfer Learning for Intelligent Triage"

_bioengineering, 2023, doi:10.3390/bioengineering10040420_

Round 1

Reviewer 1 Report

Overall, this study presents an intelligent triage system based on transfer learning to process multi-label neurological medical texts. The system uses the triage priority method to label diagnostic combinations and considers the severity of the disease to reduce the class overlapping of the data set. The BERT model is used to classify the text of the chief complaint and predict the primary diagnosis corresponding to it. The results show that the system can significantly improve the accuracy of triage and improve the patient's medical experience. The study is well-structured and provides a comprehensive overview of the intelligent triage system. The authors have successfully addressed the problems faced by patients in large hospitals and proposed a solution that could significantly improve the accuracy of triage. The use of transfer learning and the triage priority method to label diagnostic combinations is commendable and adds value to the system. Additionally, the use of the BERT model and the composite loss function has been effective in overcoming data imbalance and improving the accuracy of classification. The experimental results have been presented clearly and provide evidence of the effectiveness of the proposed system. However, it would have been useful to see more analysis and discussion of the limitations of the system and potential areas for improvement. The study only focuses on neurological medical texts, and it would be interesting to see how the system performs on other medical specialties.

Explain all the notations in the equations.

Improve the quality of Figure 1

Accuracy is considered as a performance metric but its corresponding explanation is not given in Section 4.1

What strategy is used to select the hyperparameters?

What are the limitations of the study?

System design should be explain more detail with clear explanation

Why there are no references added?

Provide dataset sources.

Is this kind of study not attempted yet? As there are lot of BERT applications in healthcare industry.

Table 2 and Table 3. represents which dataset?

Overall presentation should be improved

Reviewer 2 Report

·        The first paragraph in the introduction section should be cited

·        The structure of the remaining part of the article should be summarized after the contribution in the introduction section.

·        The research issue and objectives can be highlighted in a separate section

·        The authors should review related works in this field in a separate section and tell us the gaps discovered in the reviewed literature

·        Line 180 has an equation labelled 1; each variable should be defined accordingly. The same is true for equation 2 in line 192.

·        The dataset used in this work should be discussed, stating the attributes and even the link to the repository

·        The dataset size, collection process, and range are not mentioned.

·        The variables used in equations 5 to 10 should be defined

·        How does the proposed work overcome the problem of model overfitting

·        The conclusion section should also be included where the study will be conducted, and future works should also be suggested.

·        Overall, the English language and presentation style should be improved significantly. There contained a lot of grammatical errors and typos.

·        Some of the citations do not correlate with the references; for instance, in line 233, the authors cited Vijay [23] but referenced 'Garla, V.N.; Brandt, C. Ontology-guided feature engineering for clinical text classification. Journal of biomedical informatics 2012, 45, 992–998.'

·        It is not possible to have two citations for such as Ghulam et al. [24], [25] you have line 237, and it is even noticed that the references 24 and 25 do not have either of the authors bearing Ghulam as follows

24. Mujtaba, G.; Shuib, L.; Raj, R.G.; Al-Garadi, M.A.; Rajandram, R.; Shaikh, K. Hierarchical text classification of autopsy reports to determine mod and cod through term-based and concepts-based features. In Proceedings of the Industrial Conference on Data Mining. Springer, 2017, pp. 209–222.
25. Hastie, T.; Tibshirani, R.; Friedman, J. Overview of supervised learning. In The elements of statistical learning; Springer, 2009; pp. 9–41.

·        Check line 245; the same error still exists, so I will suggest the authors check their citations well and make sure that both kinds of literature cited are the exact literature referenced

·        It was also noticed that only one paper was used and cited in 2022, 2021 and 2020, which is like less than 5% of the literature used is recent, which is not too good enough as it does not make the paper to be up to date. Therefore the authors should review the following documents relating to the article to make the study an up-to-date one for the readers. To make your analysis more robust and interesting to readers, these are a  few kinds of literature relating to your research that you can read, cite, and reference. This will make the study more relevant and UpToDate in the research field since the literature will be published in 2022 and 2023.

1.     Lv, Z., Yu, Z., Xie, S., & Alamri, A. (2022). Deep Learning-Based Smart Predictive Evaluation for Interactive Multimedia-Enabled Smart Healthcare. ACM Trans. Multimedia Comput. Commun. Appl., 18(1s). doi: 10.1145/3468506

2.     Diao, Dan, et al. "Bayes Conditional Probability-Based Causation Analysis between Gestational Diabetes Mellitus (GDM) and Pregnancy-Induced Hypertension (PIH): A Statistic Case Study in Harbin, China." Journal of Diabetes Research (2022). Doi: 10.1155/2022/2590415

3.     Luan, D., Liu, A., Wang, X., Xie, Y., Wu, Z.,... Zhang, W. (2022). Robust Two-Stage Location Allocation for Emergency Temporary Blood Supply in Postdisaster. Discrete dynamics in nature and society, 2022. doi: 10.1155/2022/6184170

4.     Xie, B., Li, S., Li, M., Liu, C. H., Huang, G.,... Wang, G. (2023). SePiCo: Semantic-Guided Pixel Contrast for Domain Adaptive Semantic Segmentation. IEEE Transactions on Pattern Analysis and Machine Intelligence, 1-17. doi: 10.1109/TPAMI.2023.3237740

5.     Xie, X., Xie, B., Xiong, D., Hou, M., Zuo, J., Wei, G.,... Chevallier, J. (2022). New theoretical ISM-K2 Bayesian network model for evaluating vaccination effectiveness. Journal of Ambient Intelligence and Humanized Computing. doi: 10.1007/s12652-022-04199-9

6.     Ban Y, Wang Y, Liu S, Yang B, Liu M, Yin L, Zheng W. 2D/3D Multimode Medical Image Alignment Based on Spatial Histograms. Applied Sciences. 2022; 12(16):8261. https://doi.org/10.3390/app12168261

7.     Qin X, Ban Y, Wu P, Yang B, Liu S, Yin L, Liu M, Zheng W. Improved Image Fusion Method Based on Sparse Decomposition. Electronics. 2022; 11(15):2321. https://doi.org/10.3390/electronics11152321

8.     Liu, H., Liu, M., Li, D., Zheng, W., Yin, L.,... Wang, R. (2022). Recent Advances in Pulse-Coupled Neural Networks with Applications in Image Processing. Electronics, 11(20). doi: 10.3390/electronics11203264

9.     Xue, F., Cheng, J., Liu, Y., Cheng, C., Zhang, M., Sui, W.,... Zhang, C. (2022). Cardiomyocyte-specific knockout of ADAM17 ameliorates left ventricular remodeling and function in diabetic cardiomyopathy of mice. Signal Transduction and Targeted Therapy, 7(1), 259. doi: 10.1038/s41392-022-01054-3

Reviewer 3 Report

I suggest removing ‘A New’ from the manuscript title. ‘and subspecialty Settings of large hospitals’, ‘are common Modern hospitals’, etc. – why initial capitalization? ‘triage personnel is the " standard " of the triage’. - Say directly the words you mean, not in quotes, as their meaning is thus unclear (sometimes the opposite one is got). ‘Based on literature search, many scholars have studied machine learning-based development methods for building intelligent triage systems or medical text classification systems.’ – but no sources are provided. Replace it/they with the proper words to avoid confusion. ‘It uses text classification methods’. ‘and other professional groups, ward 21’ – ward 21? Some paragraphs are way too short. Some large passages are not at all substantiated. Most of the cited sources are not from peer-reviewed journals. More development and depth of the methodology and analysis are needed. The manuscript will benefit from further discussion of key concepts and methodological criteria in order to offer a better articulation between theory and data. The main contributions of the paper should be presented as part of the empirical discussions or critical assessment on the core research outcomes. The discussions require more structure and there is a need of offering a clear assessment of reviewed literature. You should compare your results with others in terms of concrete data for better research integrative value. Conclusion needs to be rewritten so that important results are brought out along with their interpretation, comparison with earlier studies, and implications in a more integrated fashion.

The manuscript has a low integrative value in hot emerging topics on intelligent triage systems. Here are some recent suggestions:

Lăzăroiu, G. (2022). “Immersive Virtual Reality Technologies, 3D Data Modeling and Simulation Tools, and Artificial Intelligence-based Diagnostic Algorithms on Metaverse Medical Platforms,” American Journal of Medical Research 9(2): 57–72. doi: 10.22381/ajmr9220224.

Nica, E. (2022). “Virtual Healthcare Technologies and Consultation Systems, Smart Operating Rooms, and Remote Sensing Data Fusion Algorithms in the Medical Metaverse,” American Journal of Medical Research 9(2): 105–120. doi: 10.22381/ajmr9220227.

Round 2

Reviewer 1 Report

Accept in present form

Reviewer 3 Report

This revised version can be published.